# UAV LiDAR Based Approach for the Detection and Interpretation of Archaeological Micro Topography under Canopy—The Rediscovery of Perticara (Basilicata, Italy)

Nicola Masini [1], Nicodemo Abate [1,*], Fabrizio Terenzio Gizzi [1], Valentino Vitale [1], Antonio Minervino Amodio [1], Maria Sileo [1], Marilisa Biscione [1], Rosa Lasaponara [2], Mario Bentivenga [3] and Francesco Cavalcante [2]

1  Consiglio Nazionale delle Ricerche, Istituto di Scienze del Patrimonio Culturale, Sede di Potenza, 85050 Potenza, Italy
2  Consiglio Nazionale delle Ricerche, Istituto di Metodologie per l'Analisi Ambientale (IMAA), 85050 Potenza, Italy
3  Dipartimento di Scienze, University of Basilicata, Campus Macchia Romana, 85100 Potenza, Italy
*  Correspondence: nicodemo.abate@ispc.cnr.it; Tel.: +39-333-336-6056

**Abstract:** This paper deals with a UAV LiDAR methodological approach for the identification and extraction of archaeological features under canopy in hilly Mediterranean environments, characterized by complex topography and strong erosion. The presence of trees and undergrowth makes the reconnaissance of archaeological features and remains very difficult, while the erosion, increased by slope, tends to adversely affect the microtopographical features of potential archaeological interest, thus making them hardly identifiable. For the purpose of our investigations, a UAV LiDAR survey has been carried out at Perticara (located in Basilicata southern Italy), an abandoned medieval village located in a geologically fragile area, characterized by complex topography, strong erosion, and a dense forest cover. All of these characteristics pose serious challenge issues and make this site particularly significant and attractive for the setting and testing of an optimal LiDAR-based approach to analyze hilly forested regions searching for subtle archaeological features. The LiDAR based investigations were based on three steps: (i) field data acquisition and data pre-processing, (ii) data post-processing, and (iii) semi-automatic feature extraction method based on machine learning and local statistics. The results obtained from the LiDAR based analyses (successfully confirmed by the field survey) made it possible to identify the lost medieval village that represents an emblematic case of settlement abandoned during the crisis of the late Middle Ages that affected most regions in southern Italy.

**Keywords:** LiDAR; UAV; landslides; deserted villages in the Middle Ages; machine learning

## 1. Introduction

Archaeological heritage in woodland is undoubtedly protected from the destructive effect of modern anthropogenic activities by the presence of tree cover, which, at the same time, prevents knowledge of them and makes investigations difficult and time consuming.

The tree cover makes geophysical prospection and excavations almost impossible and the use of remote sensing based on optical imagery quite ineffective. In these conditions, LiDAR is the only tool that enables us to "filter out" the canopy to reveal archaeological remains and microtopographical changes of cultural interest. A LiDAR scanner, mounted on aerial platforms, including unmanned aerial vehicles (UAVs), sends hundreds of thousands of pulses of light toward the area to be investigated. Most of them are reflected off the forest canopy and a few reach the ground and are reflected back through the canopy. Recording how long it takes the light to return to the scanner produces a point cloud.

Over the past two decades, LiDAR has found increasing popularity in archaeology and has opened new perspectives in the study of the human past, revolutionizing the domain of surveying to capture and depict archaeological features under canopy [1–11]. The popularity of this approach in the archaeological field is such that it has led experts to create workflows and tools for archaeology that are different from approaches used in other disciplines, as described in [12,13]. Moreover, numerous studies also adopted a standard approach, consisting of: (i) raw data acquisition and processing [7,14,15], (ii) point cloud processing and post-processing [14,16], (iii) archaeological interpretation phase [12,16,17], and (iv) dissemination [4,12,15,17].

The study of abandoned medieval settlements in highland areas is one of the fields of archaeological research that can greatly benefit from the use of LiDAR technology [5]. They are the result of "social desertification" of vast territories in Europe since the first decade of the 14th century, characterized by a demographic decline [18–22] occurring after four centuries of prosperity (from the 10th to 13th century) and population growth [19].

As a whole, there are two main reasons why LiDAR lends itself well to the survey and study of abandoned or lost medieval settlements:

1) Their location in hilly heights (in most cases for defensive reasons), especially in southern Europe, hence the need to discriminate anthropic topographic and microtopographic features from those of geomorphological nature [23,24];

2) The forest/vegetation cover that generally tends to hide a large part of these settlements, hence the need to filter out the point clouds of the vegetation to reveal the archaeological features.

These challenging conditions can be faced using:

i. A LiDAR survey with a very high density of points that typically can be obtained by UAV;

ii. Point cloud processing approaches devised for archaeological micro-relief features that are generally very subtle and, therefore, could be completely filtered out (mistaken for low vegetation) [13];

iii. Effective enhancement using digital terrain models and feature extraction methods to facilitate and improve the archaeological interpretation.

Significant advances in and from LiDAR applications have been obtained in the last decade including: (i) DTM (Digital Terrain Model), DFM (Digital Feature Model), or NVS (Non-Vegetated Surface) [12,13,25] visualization enhancement techniques, such as Sky View Factor [26], Local Relief Model [27], Openness [28], whose performances are strongly dependent on local conditions and generally evaluated subjectively through visual inspections; (ii) supervised and unsupervised classification, such as Object-Based Image Analysis (OBIA), Machine-Learning classification (ML), and Deep-Learning classification (DL) [4,29–32].

This paper presents a three-step methodological approach, based on (i) field data acquisition, (ii) data pre-processing and data post-processing aimed at data enhancement, and (iii) automatic feature extraction, devised to identify subtle archaeological microtopography under canopy in Mediterranean environments.

For this purpose, a UAV LiDAR survey has been carried out at Perticara (located in Basilicata, Southern Italy), an abandoned medieval village located in a geologically fragile area characterized by complex topography and strong erosion and covered by a dense vegetation canopy. All of these characteristics pose seriously challenging issues for the identification and extraction of the subtle microtopography of archaeological interest and make this site particularly significant and attractive for the setting and testing of LiDAR-based automatic chain processing.

From an archaeological point of view, the investigations were performed both to detect and to spatially characterize the urban shape of the site and to provide information to understand the potential causes that determined its abandonment.

From a technological and methodological point of view, the aim is to evaluate the potential of UAV LIDAR coupled with machine learning in identifying and extracting archaeological features, including not only wall remains (easy to identify) but also, and above all, subtle proxy indicators related to microtopographical variations under dense canopy.

## 2. Material and Method

### 2.1. Study Area: Historical and Archaeological Setting

The study area is the abandoned village of Perticara settled on a hill overlooking the Sauro Valley south of Basilicata, a region situated between Apulia and Campania (in Southern Italy).

Basilicata, as many other regions of southern Europe, went through a period of profound social, territorial, and economic reorganization, accompanied by a demographic decrease [33]. Many villages and rural settlements disappeared and the surviving population moved to nearby towns or villages. In particular, from 1277 to 1447 approximately 30% of the villages in Basilicata were abandoned for reasons that are still not entirely clear today. This phenomenon was accompanied by a population decline of 15% from 1277 to 1320 and of 4% from 1320 to 1447 [5].

The Sauro valley is characterized by an intense human presence from the proto-history to the Middle Ages; it is archeologically documented. In particular, there are several sites that have been permanently inhabited from Late Antiquity to the Middle Ages, such as Torri, Cornito (currently called Corleto), and the recently investigated "Eremita" site [34]. They represent different kinds of settlement, from the *vicus* (houses and lands in the same settlement, developed during Late Antiquity) to the *castrum* (medieval fortified village) (Figure 1).

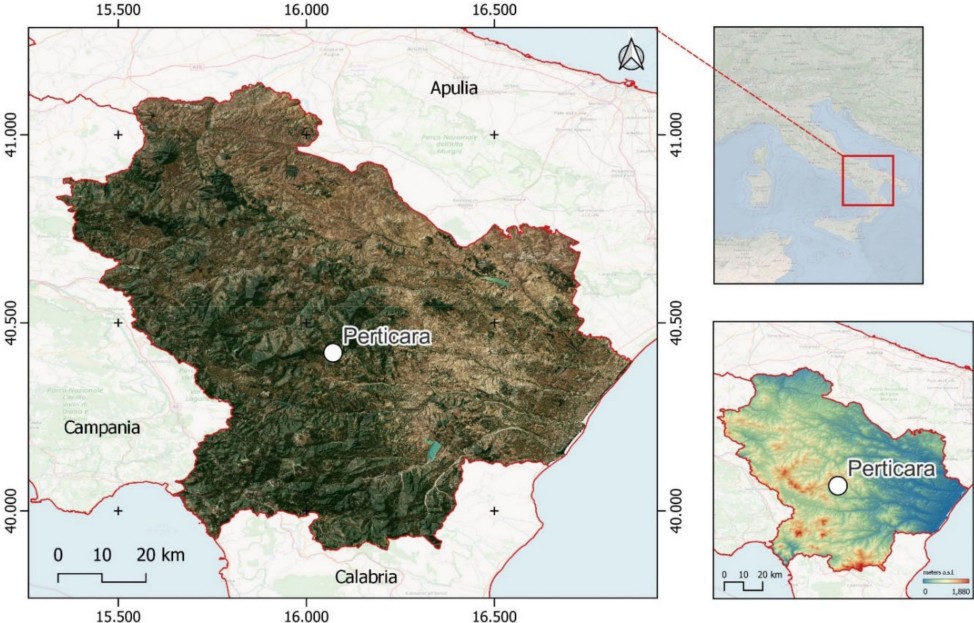

**Figure 1.** Geographical location of the site of the abandoned village of Perticara (Coordinate Reference System WGS 84 EPSG::4326).

The archaeological research in this area has brought to light numerous findings—dated from the classical age to the modern age—related to works of regimentation of meteoric and spring waters, designed and used both to (i) rationalize the use of water resources and (ii) to protect the settlements—with the related agricultural and pastoral activities— from the frequent landslides affecting the site. The attention paid to the management of hydrogeological risk allowed the foundation and development of some settlements in these geologically unstable areas. The most important are: (i) Torri, a seat of the diocese in the 12th century, located on a plateau to control the confluence of the Piscone torrent with the

river Sauro (damaged by a landslide in 2003); (ii) Perticara, named *Castrum Perticarii* in the documents of the 12th century.

Perticara was founded between the 11th and 12th centuries. It had a period of great demographic development between the mid-12th century and the 13th century. In particular, according to the Angevin tax register dating back to 1277, Perticara paid 240 "fuochi" or hearthstone (about 1200 inhabitants), more than twice as much as the neighboring centers of Corleto and Guardia (101 and 100). In 1320, the number of hearthstones halved. The decline of the village continued without interruption until it disappeared in the 15th century, as evidenced by the taxation of 1447 that no longer included Perticara among taxed inhabited settlements. The causes of this abandonment—that probably occurred between the end of the 14th century and the beginning of the 15th century—are various; they are linked to global and local factors, as in other areas of the region. In Basilicata, many rural settlements came into crisis due to the tax burden, epidemics, pestilences, and widespread conflicts in the first decades of the Angevin age [35]. Demographic decline and settlement abandonment are frequent, especially in the case of destructive events such as an earthquake or a landslide [36] that probably caused the decline of Perticara.

The historical reconstruction of the agricultural landscape and land use, the approximate data of the population (taken from the tax registers), and the archaeological and architectural evidence allow us to add information to the general and local framework and to interpret them.

### 2.2. Geological and Geomorphological Setting

The site is located between the axial and the frontal area of the southern Apennine Mountains, 5 km northeast of Corleto Perticara village (Figure 2a).

The southern Apennines consist of a fold-and-thrust belt developed from the Upper Oligocene–Lower Miocene boundary onwards as a result of the tectonic accretion wedge towards NE of different Meso–Cenozoic paleodomains [37] (Figure 2b).

Perticara is located along the Caperrino Ridge. This morphostructure is 12 km long and NNW-SSE oriented. It separates the Fiumara di Gorgoglione drainage basin in the north-east from the Fiumarella di Corleto drainage basin in the south-western area. The geomorphological features of the area are closely related to the outcropping units. In particular, lithology and bedding attitude strictly control the acclivity of the slope. Slopes with high acclivity widely occur in the eastern sectors. Locally, the slope gradient is reduced considerably only in areas where clay deposits such as varicolored clays and shales crop out. Along the basin slopes, different geomorphological features such as trenches and morphological steps are very common. The landslides recognized in the study area have been classified as complex landslides [38]. In particular, the landslides occurring in the study area are controlled by a large number of factors, some of them often acting in concert. Among these, bedding attitude, tectonic structures, the thickness of the various lithological units, and the lithology play a fundamental rule. Differences in permeability and competence of the various rock types, which are strictly linked with compositional variations and/or grain size, are other important factors to take into account. In addition, the landslides detected were triggered in the past by abundant precipitation. Land sliding is also conditioned by topographic parameters, such as the slope gradient and the exposure. In particular, this latter parameter controls soil moisture and consequently the amount of vegetation cover. Based on a detailed field survey, supported by the analysis of stereoscopic aerial photographs, different landslide types, including earthflows, complex landslides, and falls, have been recognized and mapped (Figure 3). In addition to landslides, areas affected by slow soil movements such as solifluction have been detected. In particular, the landslide that led to the abandonment of the Perticara site started in the south-eastern part of the crest of Caperrino Mount and diverted the Fiumarella di Corleto path downstream. The landslide was classified as a large earthflow and is currently considered dormant [39].

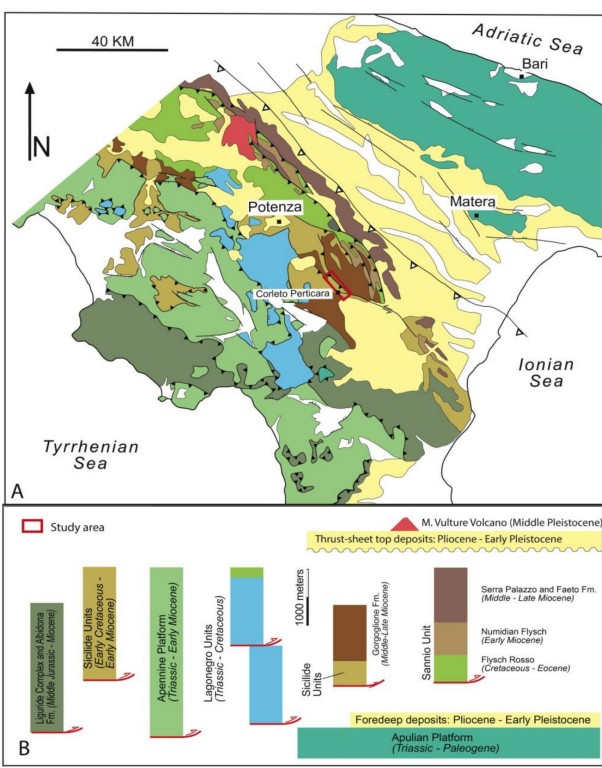

**Figure 2.** (**A**) Geological sketch map of the southern Apennines, (**B**) relationship between different stratigraphic–structural units ([40], modified).

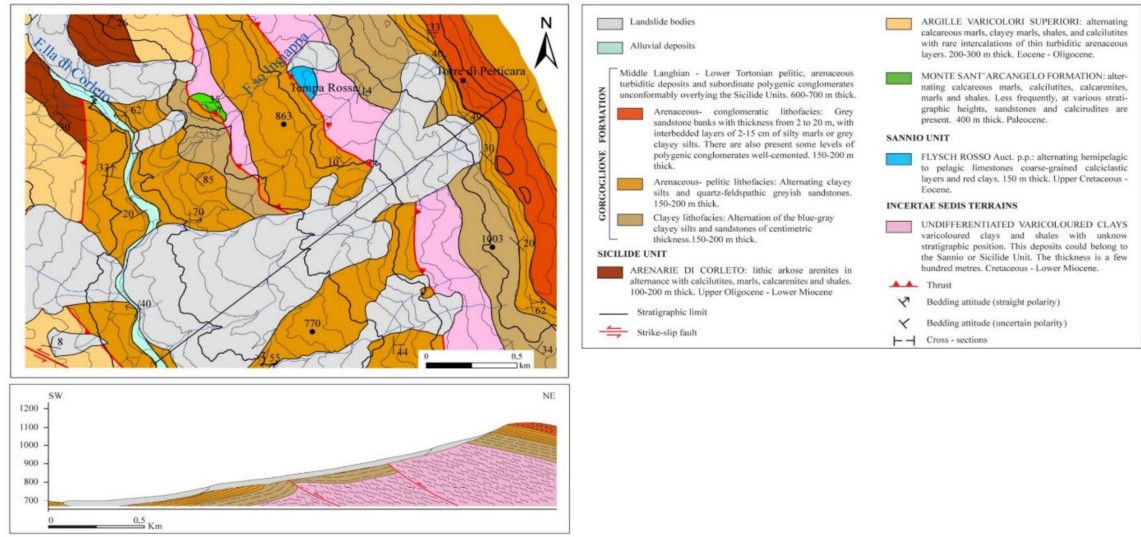

**Figure 3.** Geomorphological setting of the Perticara site.

### 2.3. Methods

The methodology used for the analysis at the Perticara site can be summarized in three different and separate phases: (i) field data acquisition and pre-processing, (ii) data post-processing, and (iii) automatic feature extraction (Figure 4).

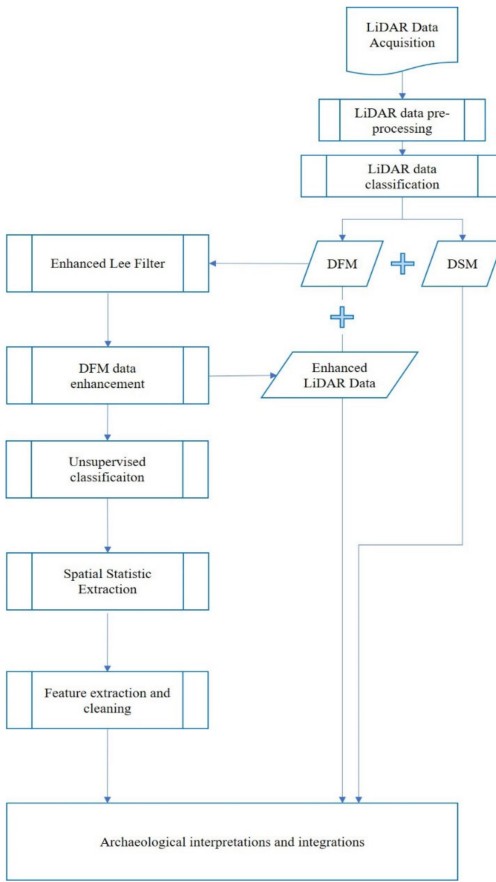

**Figure 4.** LiDAR processing flowchart.

### 2.3.1. Field Data Acquisition and Data Processing

The LiDAR survey at the Perticara site was carried out using a 5-echo Riegl MiniVux-3 LiDAR (RIEGL Laser Measurement Systems GmbH, Austria), equipped with a GNSS PPK positioning system, used as a payload on a DJI Matrice 600 drone.

The LiDAR acquisition covered a useful area, i.e., free of noise due to beam scattering, of 10,645 ha. The flight was conducted at an altitude of 70 m a.g.l. (above ground level), with a lateral spacing between strips of 20 m, at a constant speed of 3 m/s, and a 120 degree FOV (Field of View), in a double acquisition grid mode, using the UgCS pro v.4.6520 software (SPH Engineering, Latvia), using the DEM (Digital Elevation Model) provided by Tinitaly (http://tinitaly.pi.ingv.it/) for the Italian peninsula [41–44] (Figure 5a,b). The acquisition was conducted in a double grid as it was considered by the authors advantageous compared with a single acquisition (Figure 5c–f). In fact, by having a LiDAR operating on a drone and not an airplane/helicopter, it was very easy to set up the second flight plan, with an expenditure of time and resources of a few tens of minutes.

The flight was conducted on 12 December 2021, during a period in which there is less undergrowth and less foliage in the trees with respect to other months, to the advantage of a greater amount of ground points. The local vegetation consists of oak, downy oaks, broom, and grass.

The data acquisition was then followed by the data pre-processing phase, as described by Riegl for this instrument. The pre-processing phases to pass from the data acquired from the LiDAR to the raw georeferenced point cloud were: (i) acquisition of the RINEX GNSS data coming from the fixed stations located on the Italian peninsula; (ii) correction of the route acquired by the PPK antenna on the basis of the data from the fixed stations in the Applanix POSPac UAV software v.8.7 (Applanix, Richmond Hill, Ontario); (iii) use of Riegl's RiPROCESS v.1.9 suite for the creation of the point cloud in the WGS 84 UTM 33 N system. Riegl's Riprocess software allows point clouds to be generated from LiDAR

data acquired at multiple times (several acquisitions) by reprocessing the data as if they were from a single flight to obtain a single point cloud. The software uses the Project Merge Wizard and RiPRECISION commands to merge the two flights by (i) setting the roto-translation matrices to a raw matrix and then re-processing a valid one for both, (ii) generating the point cloud using the data from the two scans, and (iii) refining the global alignment of the data.

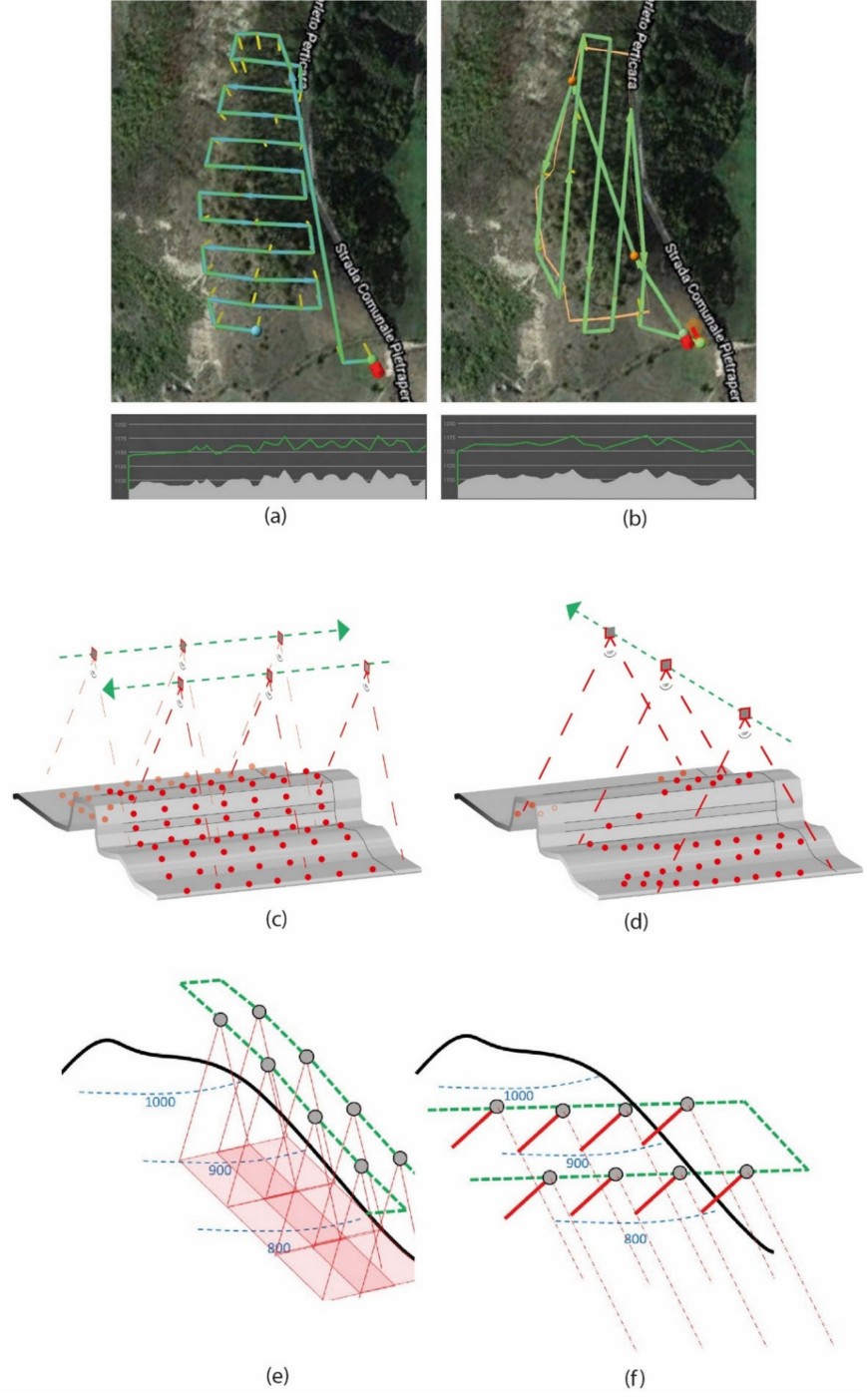

**Figure 5.** Representation of the flight route performed for the LiDAR survey: (**a**) first route and elevation profile, (**b**) second route and elevation profile, (**c**,**d**) schematization of the advantage of the methodology for structures/topographical microreliefs; (**e**,**f**) schematization of the advantage of the methodology on sloped landscapes.

According to the LiDAR specifications, under ideal conditions, each individual acquisition strip generated a point cloud with a density of 142 points/m$^2$. However, the acquisition took place in dual grid mode, which affected the density of points per m$^2$. The density was inhomogeneous due to a number of determining factors such as (i) sum of the points/m$^2$ of the two individual point clouds generated by the individual acquisitions, (ii) drone speed subject to microdelays/accelerations due to wind, and (iii) morphological discontinuity of the recorded subject.

### 2.3.2. Cloud Point Processing and Automatic Feature Extraction

Once the georeferenced point cloud was obtained by LiDAR processing (see Section 2.3.1), procedures for point classification were prepared. Several methods for classifying ground points from off-ground points can be found in the literature [29,30,45–52]. The extraction of the ground profile from the point cloud was achieved using Global Mapper® v.22.1 software. Global Mapper® (Blue Marble Geographics, Maine, U.S.A.) uses a hybrid filter type (BMHF) [45].

The classification operation of the point cloud to obtain the DFM (terminology chosen in accordance with [13] as it indicates both the digital terrain model and the features of archaeological interest embedded in it or completely above ground) were: (i) classification of ground points from off-ground points using the automatic classification algorithm in the software; (ii) removal of noise points (i.e., points too far from the points classified as ground). Finally, the classified point cloud was exported. The classified point cloud, for study interest classes only, had a varying density of points per m$^2$ from a minimum in areas with high vegetation (0 to 30 points/m$^2$ approx.), to areas with less dense vegetation (80–150 points/m$^2$ approx.), to a maximum in bare areas (600–800 points/m$^2$ approx.).

The point cloud was then subjected to a Spatially Resampling Interpolation using the open-source software Cloud Compare v.2.12.4 to obtain a cloud with a constant point density, set with a spacing GSD (Ground Sample Distance) of 0.02 m, using the rasterize tool. The aim of this tool is to convert the point cloud into a 2.5D grid that can be re-exported as a new point cloud, mesh, or raster (georeferenced) [53]. This task can also be conducted using the function LasThin in the LASTools software (Rapidlasso GmbH, Germany), often used for classifying LiDAR data in archaeology [12,13]. The DFM was then created and exported using the same command, with a cell of 0.02 × 0.02 m (Figure 6).

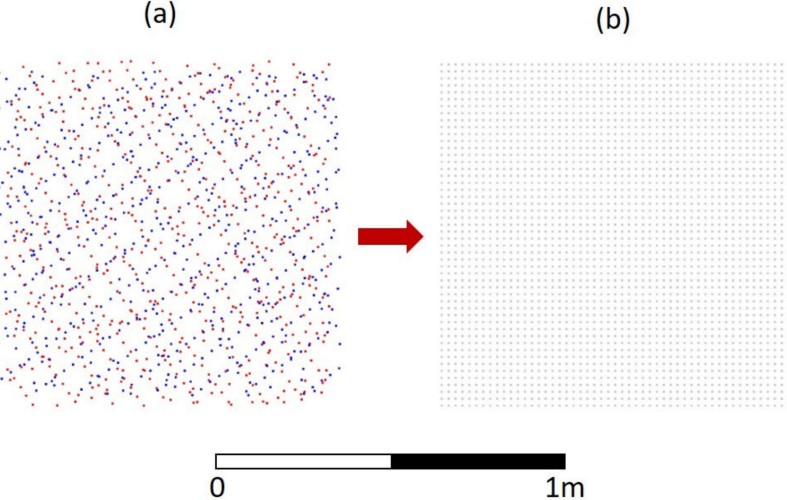

**Figure 6.** Comparison of (**a**) point cloud created by merging data from the two LiDAR acquisitions, and (**b**) point cloud generated by Cloud Compare.

The DFM obtained was then subjected to several operations to improve the rendering and visibility of the archaeological features. The operations were of two types: (i) noise reduction of the DFM, according to the methodology already proposed in [4,29,54];

(ii) creation of DFM derived from different visualization techniques, as proposed for archaeological studies in [1,5,26,55–57].

Noise and speckle reduction was done using the GRASS GIS operator, in the QGIS software. The algorithm used was an enhanced Lee filter [58]. Lee filter reduces noise by applying a spatial filter to each pixel and is based on the analysis of local statistics calculated in a square window. The value of the pixel in the center of the window (set to $9 \times 9$ in the present study) is calculated as the mean or weighted average of the neighboring pixels. The enhanced Lee filter is an improved version of the Lee filter that not only reduces noise but also preserves the detail and sharpness of the original image [59]. The value of the beveled pixel is then calculated as (1):

$$
\begin{aligned}
& L_M \text{ for } C_I \leq C_U \\
& L_M * K + P_C * (1 - K) \text{ for } C_U < C_I < C_{max} \\
& P_C \text{ for } C_I \geq C_{max}
\end{aligned}
\tag{1}
$$

where

I.　　$L_M$ is the Local Mean of filter window;
II.　　$C_U = \frac{1}{\sqrt{Nlooks}}$ is the noise variation coefficient;
III.　　$C_{max} = \sqrt{\frac{1+2}{NLooks}}$ is the maximum noise variation coefficient;
IV.　　$C_I = \frac{SD}{L_M}$ is the image variation coefficient;
V.　　$K = e^{(-D(C_I - C_U)/(C_{max} - C_I))}$;
VI.　　$P_C$ is the Center Pixel value of window;
VII.　　SD is the Standard Deviation in filter window;
VIII.　　NLooks is the Number of Looks;
IX.　　D is the Damping factor.

After filtering with the enhanced Lee filter, the second operation performed was the enhancement of DFM, creating derivatives based on a number of visualization techniques. These techniques are generally based on how the illumination interacts with the points in the DFM as discussed in [5,26,57,60]. For this study, the open source tool RVT (Relief Visualization Toolbox) developed by Kokalj et al. [57] was used in order to create useful derivatives for site analysis (Table 1).

**Table 1.** Derivatives based on visualization techniques.

| Visualization Method | Parameters |
| --- | --- |
| Analytical Hillshading | Sun azimuth (deg): 315; Sun elevation angle (deg): 35 |
| Hillshading from Multiple Directions | Number of directions: 16; Sun elevation angle (deg): 35 |
| PCA of Hillshading | Number of components to save: 3 |
| Slope Gradient | No parameters required |
| Simple Local Relief Model | Radius for trend assessment (pixel): 20 |
| Sky-View Factor | Number of search directions: 16; search radius (pixel): 20 |
| Openness Positive | Number of search directions: 16; search radius (pixel): 20 |
| Openness Negative | Number of search directions: 16; search radius (pixel): 20 |
| Archaeological (VAT) | Used preset: general |

The data thus produced were then used for the classification of features of interest in the reconstruction of the archaeological context.

### 2.3.3. A Machine Learning-Based Approach for a Semi-Automatic Feature Extraction

The process of classification to extract features of archaeological interest is based on the following steps: (i) selection of data to be classified; (ii) choice of the classifier; (iii) preparation of data for the classification; (iv) classification run; (v) extraction of local statistics;

(vi) feature identification and cleaning. The classification operations were automatic (unsupervised classification); however, the second stages of operations, namely segmentation, cleaning and identification, were supervised.

The data used for the classification were those produced by the DFM visualization enhancement processing as described in Section 2.3.2 specifically: (i) Hillshading from Multiple Directions (HS); (ii) Slope Gradient (Slope); (iii) Simple Local Relief Model (SLRM); (iv) Sky-View Factor (SVF); (v) Anisotropic Sky-View Factor; (vi) Openness Positive (OP); (vii) Openness Negative (ON); (viii) Archaeological (VAT) derived from a blend of SVF, OP, Slope, and Analytical Hillshading.

The choice of the data to be used was then succeeded by the choice of the classifier to be used before the feature segmentation procedures. The preparatory operations for classification and ISODATA classification were done using SAGA GIS 7.8.2 operating within the QGIS environment [61].

A data normalization operation was applied before classifying the data. The goal of normalization operations is to transform all data to a similar scale in order to improve the performance of classification algorithms. There are several types of normalization, such as: (i) scaling to a range; (ii) clipping; (iii) log scaling; (iv) z-score [62–65]. The type of normalization used is scaling, which means converting feature values into a standard range (e.g., 0 to 1, −1 to 1), according to the Formula (2):

$$X^I = (X - X_{min})/(X_{max} - X_{min}) \tag{2}$$

The data thus processed were subsequently used for classification.

The operations were carried out using an unsupervised ISODATA classifier. There are several types of unsupervised classifiers used in the context of Remote Sensing studies applied to archaeology, such as (i) the Kmeans clustering algorithm and (ii) the ISODATA (Iterative Self-Organizing Data Analysis) algorithm [66–68]. Unsupervised classification algorithms classify pixels on the basis of their characteristics (e.g., spectral feature) without the need for prior training. This is optimal in the context of an archaeological study based on LiDAR data derivatives, as features of archaeological interest are not prior known [69–72]. Seven classes were used for clustering. The present study was conducted using an ISODATA-type classification.

A LISA (Local Indicator of Spatial Autocorrelation) process was then applied to the classified data via ISODATA to improve the visualization of the spatial aggregation and autocorrelation of pixels [73,74].

The function was applied to calculate the spatial autocorrelation using the indices: (i) Moran's I, (ii) Geary's C, and (iii) Getis–Ord G index [75–77]. Moran's I, Geary's C, and Getis–Ord G indices were calculated in accordance with Anselin (3, 4) [75] and Getis–Ord Formula (5) [76]:

$$I_i = Z_i \sum_j W_{ij} Z_j \tag{3}$$

$$C = \frac{(N-1) \sum_i \sum_j W_{ij} (x_i - x_j)^2}{2(\sum_i \sum_j W_{ij}) \sum_i Z_i^2} \tag{4}$$

$$G_i^* = \frac{\sum_j W_{ij}(d) x_j}{\sum_j x_j} \tag{5}$$

where (i) $Z_i$ is the deviation of the variable of interest; (ii) $W_{ij}$ is a spatial contiguity matrix with a zero diagonal and the off-diagonal non-zero elements indicating the contiguity of positions i and j; (iii) N is the number of the pixels; (iv) $x_i$ and $x_j$ are intensity in points i and j (with $i \neq j$) [77]. The spatial autocorrelation analysis outputs a new image where pixels are aggregated for their correlation in a window around a pixel, highlighting features not always immediately visible and reducing background noise (e.g., salt and pepper).

Among the created indices of local spatial autocorrelation, the Getis–Ord G index was used for the subsequent segmentation phase in order to extract features of interest

in the reconstruction of the archaeological landscape. Segmentation is a process of the Object-Based Image Analysis (OBIA) that involves the classification of features on the basis of several variables such as (i) pixel value, (ii) object shape, (iii) textural information, (iv) neighborhood analysis, as opposed to pixel-oriented classification, usually used in the analysis of LiDAR data for archaeological purposes [32,60,78–81]. For this study, segmentation was applied directly to the Getis–Ord G index, with the following parameters: (i) spatial radius of the neighborhood equal to 5; (ii) minimum region size equal to 40. Finally, the resulting vector file was further cleaned using a spatial criterion, with a threshold based on the surface area (area in m$^2$) of the individual features. The filtering scheme was carried out using a progressive threshold as follows: (i) all vectors with area < 1 m$^2$ and area > 12,000 m$^2$ were removed (considered to be scattered pixels and background noise of the source data, respectively); (ii) relying on the automatic categorization of the QGIS software for vector display, vectors with area < 3 m$^2$ were removed because they are not considered useful for archaeological purposes.

All the data thus produced were then analyzed and interpreted in a GIS system.

The data derived from the LiDAR acquisition were also observed by archaeologists to manually trace structures and microreliefs of possible archaeological interest identifiable on them.

Finally, in order to evaluate the accuracy of the result obtained from the automatic extraction, functions were applied to understand the overlap of the automatically extracted features and those optically identified by the archaeologists, following the example proposed by Masini et al. [4].

The method used to estimate the linear length overlap between automatically identified features and manually identified features in the different LiDAR-derived products is described by the Formula (6):

$$\mu x_i = \frac{Lx_{iFDM} - Lx_{iAEF}}{Lx_{iFDM} + Lx_{iAEF}} \tag{6}$$

where $\mu x_i$ is a modified version of the normalized visibility index proposed by Masini et al. [4], referring to the linear length overlap between the length of segments identified automatically by the automatic extraction process ($Lx_{iAEF}$) and those identified optically on the different derived LiDAR product ($Lx_{iFDM}$). The result is a number between $-1$ and 1, where values close to 0 indicate a good match while values tending toward $-1$ and 1 indicate underestimation or overestimation of the segment length by the segmentation algorithm, respectively.

The segments considered for this analysis were then divided into classes (e.g., walls, buildings, and tower) and Formula (7) was applied to calculate the weighted average normalized visibility index as proposed in Masini et al. [4].

$$\mu_{LDM} = \frac{\sum Lx_{iFDM} * \mu x_i}{\sum Lx_{iAEF}} \tag{7}$$

## 3. Results and Discussion

### 3.1. Results and Discussion: LiDAR Data, Derived LiDAR DFM, and Automatic Feature Extraction Methods

The elaborations conducted on the data acquired at the Perticara site have improved the understanding of the medieval site, as well as the morphology of the landscape in the immediate surroundings. The key elaboration in the extraction of the information was the point cloud classification. The result was a quite clean DFM, subsequently further smoothed thanks to the enhanced Lee filter (Figure 7).

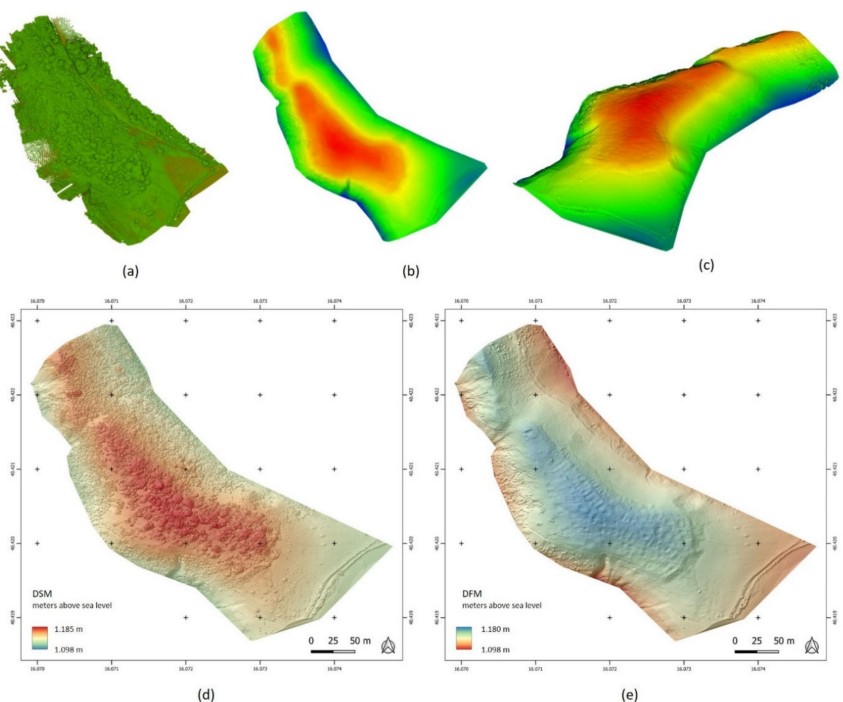

**Figure 7.** (**a**) Classified ground/non-ground point cloud; (**b**) filtered point cloud (top view); (**c**) filtered point cloud (perspective view); (**d**) Digital Surface Model (DSM), hillshading; (**e**) enhanced Lee filtered Digital Terrain Model (DFM), hillshading.

Although the DFM showed microreliefs and traces of archaeological interest, the products derived from it (see Section 2.3.2) significantly improved this visibility (Figure 8).

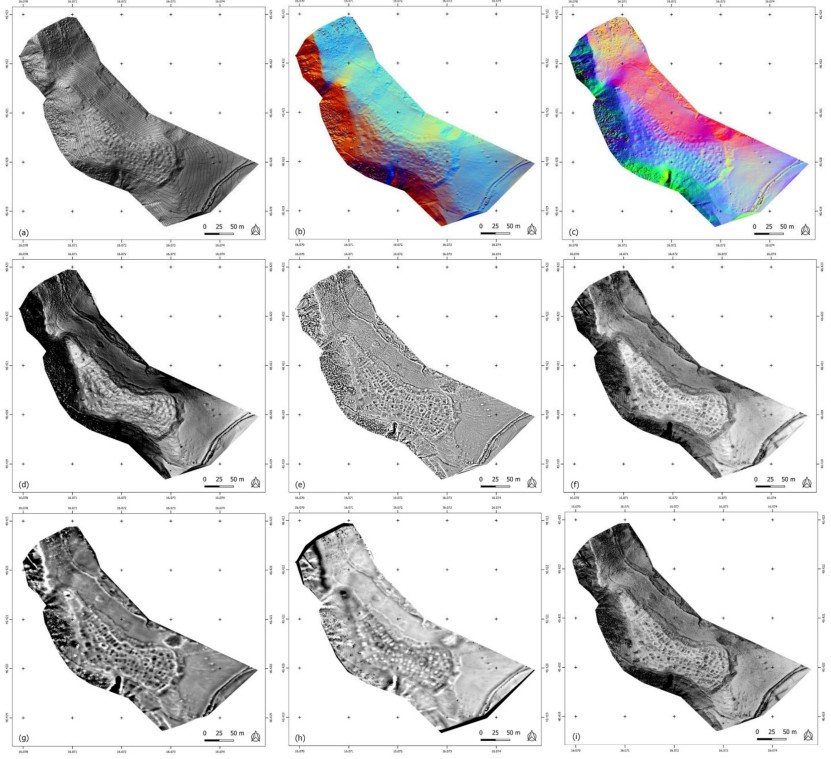

**Figure 8.** DFM derived products: (**a**) HS; (**b**) MHS; (**c**) PCA of HS; (**d**) Slope; (**e**) SLRM; (**f**) SVF; (**g**) OP; (**h**) ON; (**i**) Archaeological VAT.

Among the DFM-derived products: (i) Hillshading, Multiple Hillshading (MSH), and PCA (Principal Component Analysis) of Hillshading are among the most automatically understandable and provide the immediate overview of the ground and of the microreliefs; (ii) on the contrary, Slope, SLRM, SVF, OP, ON, and Archaeological VAT require a minimum of interpretation, although they greatly improve the visibility of microreliefs, archaeological structures, and landscape elements (e.g., landslides, quarries).

In addition, derivative products were crucial for automatic classification by unsupervised classifiers. The result of the ISODATA classification was a single output, against multiple derived products, from which only features of interest to the presented study were subsequently extracted. However, the automatic feature extraction was achieved by applying a segmentation step obtained through LISA (Local Indicators of Spatial Association) indices, of which the most useful in this particular case was the Getis–Ord G index.

An important contribution was made by the interpretation of features by archaeologists and geomorphologists using GIS, which allowed: (i) the classification of features useful for the reconstruction of the ancient context, (ii) the exclusion of false positives, and (iii) the recording of features missed by the unsupervised classification (Figure 9).

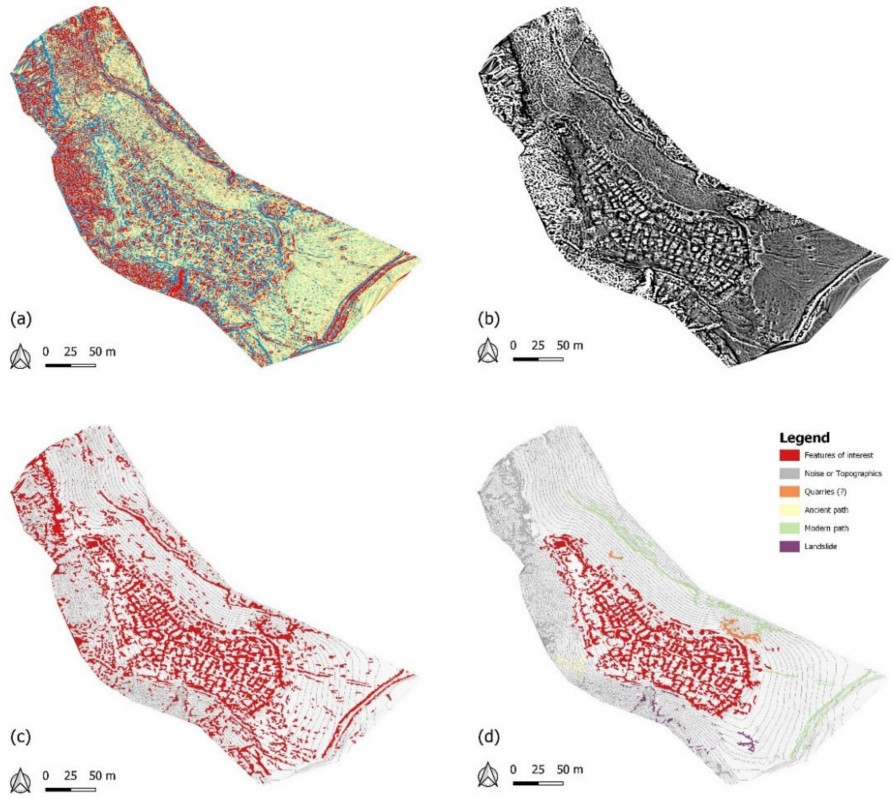

**Figure 9.** (**a**) ISODATA result; (**b**) Getis–Ord G index; (**c**) extracted features after the first cleaning operation; (**d**) archaeological and geomorphological characterization after second cleaning operation (see Section 2.3.3).

The method is inspired by Masini et al. [4] for the medieval site of Cisterna, which is developed according to the scheme: (i) LiDAR derived (i.e., SVF), (ii) LISA (Geary's C), (iii) ISODATA, and (iv) segmentation [4].

In difference to this process, the method used by this study is structured in such a way as to (i) minimize manual operations by merging the data through normalization and classification operations and (ii) reduce the salt-and-pepper effect produced by the high resolution, through LISA (Getis–Ord G index), before performing segmentation. In fact, the substantial difference between the two studies lies in the different resolution of the data, which in the case of Cisterna reaches 0.5 m/pixel while in the case of this work it is about 0.02 m/pixel. The very high resolution available in the case of the Perticara datum on the

one hand allows for a sharp image of the features and microreliefs of archaeological interest, while on the other hand it presents a large amount of microfeature and spackle, especially when subjected to Geary's C index. For this reason, the process has been modified so that pixels are clustered in the best way possible (e.g., enhanced Lee filter, classification, LISA) and the noisy effects that such high resolution can give in the case of automatic feature identification are reduced.

### 3.2. Results and Consideration about Accuracy Assessment of the Automatic Extraction Method

The results of the comparison between automatically and manually identified features, obtained as described in the methodologies, are shown in Table 2 and Figures 10 and 11.

**Table 2.** Testing for the accuracy of the automatic versus the manual extraction method. The best results are achieved for the normalized visibility index of archaeological features that are related to OP and SLRM for building feature and to SVF for walls and perimeter features.

| | xi | | L (m) | OP (m) | SLRM (m) | PCA (m) | SVF (m) | Slope (m) | VAT (m) | μOP | μSLRM | μPCA | μSVF | μSlope | μVAT |
|---|---|---|---|---|---|---|---|---|---|---|---|---|---|---|---|
| **Wall and perimeter features** | W1 | | 28.3 | 28.4 | 28 | 26.7 | 27.8 | 28.8 | 28.8 | 0.0018 | −0.0053 | −0.0291 | −0.0089 | 0.0088 | 0.0088 |
| | W2 | | 21.1 | 21.1 | 22 | 21.6 | 21.9 | 21.3 | 21 | 0.0000 | 0.0209 | 0.0117 | 0.0186 | 0.0047 | −0.0024 |
| | W3 | | 52.6 | 52.8 | 49.2 | 50.1 | 52.4 | 48.1 | 52 | 0.0019 | −0.0334 | −0.0243 | −0.0019 | −0.0447 | −0.0057 |
| | W4 | | 19.2 | 10 | 17.4 | 16.5 | 17.6 | 16.8 | 15.8 | −0.3151 | −0.0492 | −0.0756 | −0.0435 | −0.0667 | −0.0971 |
| | W5 | | 32 | 33 | 36 | 30.6 | 33.1 | 23.4 | 32.1 | 0.0154 | 0.0588 | −0.0224 | 0.0169 | −0.1552 | 0.0016 |
| | W6 | | 48.6 | 39.7 | 31.1 | 44.4 | 48 | 49 | 46.4 | −0.1008 | −0.2196 | −0.0452 | −0.0062 | 0.0041 | −0.0232 |
| | ∑Lw | μLDM | | −0.032 | −0.034 | −0.028 | −0.002 | −0.031 | −0.013 | | | | | | |
| | | | | −3.20% | −3.4% | −2.8% | −0.20% | −3.10% | −1.3% | | | | | | |
| | B1 | | 83 | 78.3 | 83.8 | 49 | 75 | 83.2 | 82 | −0.0291 | 0.0048 | −0.2576 | −0.0506 | 0.0012 | −0.0061 |
| | B2 | | 14.7 | 13.5 | 13.8 | 10.7 | 11.8 | 12.4 | 13.9 | −0.0426 | −0.0316 | −0.1575 | −0.1094 | −0.0849 | −0.0280 |
| | B3 | | 16 | 14.7 | 14.6 | 11.6 | 12.3 | 11 | 12.4 | −0.0423 | −0.0458 | −0.1594 | −0.1307 | −0.1852 | −0.1268 |
| | B4 | | 17.9 | 19.2 | 20 | 17.7 | 16.3 | 15.3 | 15.8 | 0.0350 | 0.0554 | −0.0056 | −0.0468 | −0.0783 | −0.0623 |
| | B5 | | 22 | 21.3 | 21.3 | 19.9 | 20 | 19.5 | 20.5 | −0.0162 | −0.0162 | −0.0501 | −0.0476 | −0.0602 | −0.0353 |
| | B6 | | 14 | 13.5 | 13.1 | 13.4 | 13.6 | 12.9 | 12.9 | −0.0182 | −0.0332 | −0.0219 | −0.0145 | −0.0409 | −0.0409 |
| | B7 | | 7.9 | 7.5 | 7.5 | 7.6 | 7.5 | 7.1 | 7.6 | −0.0260 | −0.0260 | −0.0194 | −0.0260 | −0.0533 | −0.0194 |
| | B8 | | 19 | 21 | 21.8 | 18.4 | 18.1 | 17.1 | 18.8 | 0.0500 | 0.0686 | −0.0160 | −0.0243 | −0.0526 | −0.0053 |
| | B9 | | 48 | 49.5 | 49.1 | 46.9 | 46.1 | 46.1 | 48.5 | 0.0154 | 0.0113 | −0.0116 | −0.0202 | −0.0202 | 0.0052 |
| | B10 | | 9.6 | 6.9 | 9.1 | 8.4 | 8.6 | 7.9 | 8.8 | −0.1636 | −0.0267 | −0.0667 | −0.0549 | −0.0971 | −0.0435 |
| **Buildings** | B11 | | 15.6 | 22.9 | 15.1 | 22 | 23.6 | 20.9 | 21.8 | 0.1896 | −0.0163 | 0.1702 | 0.2041 | 0.1452 | 0.1658 |
| | B12 | | 32.9 | 31.1 | 31.1 | 31.8 | 26.6 | 26.1 | 30.3 | −0.0281 | −0.0281 | −0.0170 | −0.1059 | −0.1153 | −0.0411 |
| | B13 | | 64.8 | 66 | 58.7 | 64.1 | 59.3 | 0 | 61.6 | 0.0092 | −0.0494 | −0.0054 | −0.0443 | −1.0000 | −0.0253 |
| | B14 | | 31.2 | 30.9 | 31.7 | 30.8 | 29.3 | 30.6 | 26.5 | −0.0048 | 0.0079 | −0.0065 | −0.0314 | −0.0097 | −0.0815 |
| | B15 | | 14.1 | 14.7 | 14.9 | 13.7 | 13.4 | 13.4 | 13.4 | 0.0208 | 0.0276 | −0.0144 | −0.0255 | −0.0255 | −0.0255 |
| | B16 | | 26.1 | 25.9 | 25.3 | 24.4 | 24.6 | 24.1 | 25.2 | −0.0038 | −0.0156 | −0.0337 | −0.0296 | −0.0398 | −0.0175 |
| | B17 | | 11.7 | 15.9 | 15.8 | 14.4 | 15.1 | 11.5 | 14.7 | 0.1522 | 0.1491 | 0.1034 | 0.1269 | −0.0086 | 0.1136 |
| | B18 | | 12 | 13 | 12.4 | 11.9 | 12 | 12 | 12.6 | 0.0400 | 0.0164 | −0.0042 | 0.0000 | 0.0000 | 0.0244 |
| | B19 | | 10.6 | 12 | 10.8 | 11 | 10.4 | 9.4 | 10.4 | 0.0619 | 0.0093 | 0.0185 | −0.0095 | −0.0600 | −0.0095 |
| | B20 | | 17.9 | 16.7 | 17.4 | 17 | 15.3 | 14.6 | 15.1 | −0.0347 | −0.0142 | −0.0258 | −0.0783 | −0.1015 | −0.0848 |
| | ∑Lw | μLDM | | 0.0088 | −0.0002 | −0.0332 | −0.0259 | −0.0257 | −0.0135 | | | | | | |
| | | | | 0.88% | −0.02% | −3.32% | −2.59% | −2.57% | −1.35% | | | | | | |
| **Tower** | T1 | | 32.2 | 28.7 | 28.5 | 36.1 | 35.8 | 36 | 35.4 | −0.0575 | −0.0610 | 0.0571 | 0.0529 | 0.0557 | 0.0473 |
| | ∑Lw | μLDM | | −0.0512 | −0.0540 | 0.0640 | 0.0589 | 0.0623 | 0.0520 | | | | | | |
| | | | | −5.12% | −5.4% | 6.4% | 5.89% | 6.23% | 5.2% | | | | | | |

Automatically extracted data were compared with those identified by archaeologists on LiDAR derivatives (Figure 10a,b).

The testing for the accuracy of the automatic versus the manual extraction method provided two interesting results. The first is based on the length of segments identified. As shown in Table 2 and Figures 10 and 11, if a segment was identified simultaneously by

the automatic extraction method and the autopsy identification, the discrepancy between the two is in most cases very small. As shown in Table 2, in many cases an average overall overlap between different LiDAR derivatives greater than 90% is achieved (e.g., w1: 98%, w2: 98%, w3: 96%, b2: 95%, b6: 96%). In addition, Table 2 shows values very close to 0 (high percentage of accuracy) ranging from +1 to −1, in agreement with Formula (6). These values are negative on average and indicate that the adopted automatic extraction system tends to slightly overestimate the length of segments.

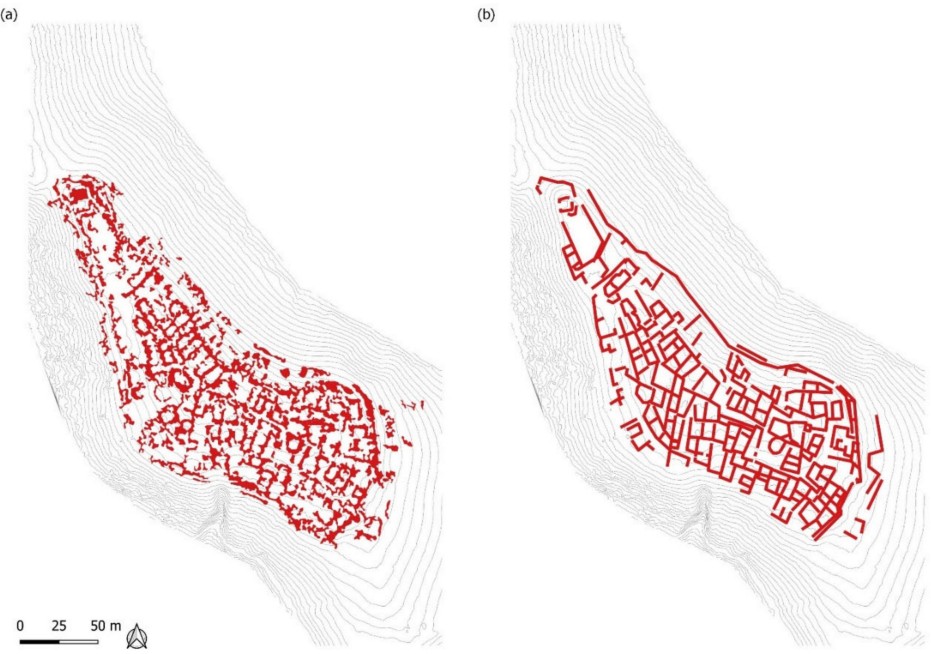

**Figure 10.** (**a**) Archaeological and geomorphological characterization after second cleaning operation; (**b**) manually identified features.

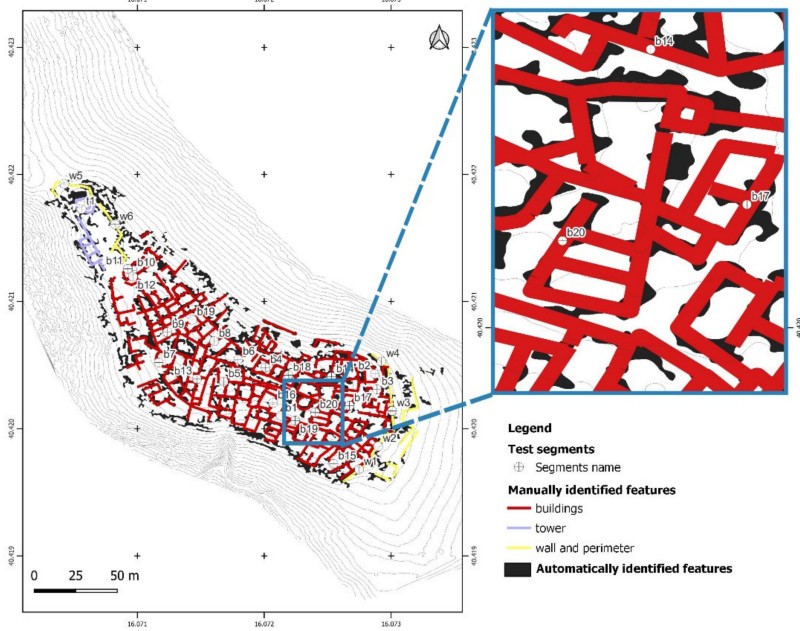

**Figure 11.** Comparison of the length of segments identified automatically and manually, according to Formula (6).

Figure 12 shows aggregated data by archaeological features (walls, buildings, and towers) and the enhancement techniques (OP, SLRM, PCA, SVF, Slope, VAT).

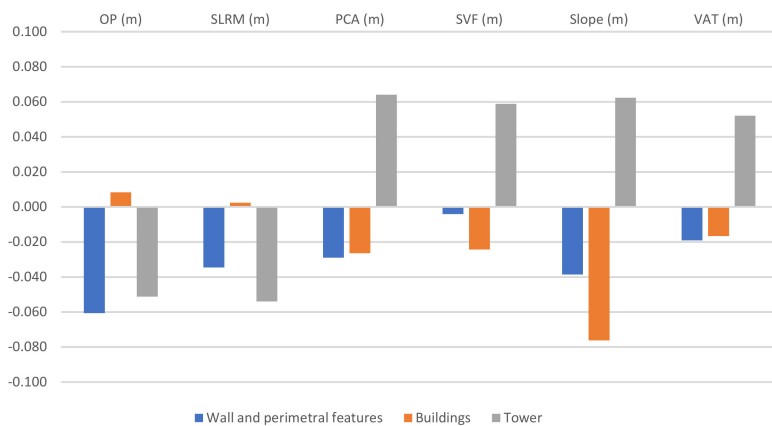

**Figure 12.** Normalized visibility index of archaeological features from LiDAR derived models (see Table 2).

Analyzing the results for the diverse archaeological features, the lowest values of the normalized visibility index ($\mu x_i$,) corresponding to an optimal matching between segments identified automatically by the automatic extraction process ($Lx_{iAEF}$) and those identified optically on the different derived LiDAR product $Lx_{iFDM}$) are recorded for the microrelief relating to the buildings.

Analyzing the data for the different enhancement techniques and observing the building features, the best results are recorded for OP and SLRM (0.88%, 0.02%, respectively).

The highest values of the normalized visibility index (related to worse results in terms of matching between extracted features and optically identified features) are recorded for the tower (average values ranging 5.1% to 6.4%), the only feature clearly visible on the ground as walled remains of it are preserved. The reason is given by the fact that on the sides of the wall structures there is abundant collapse material, increasing the size of the automatically extracted segments.

However, although the overlap of features identified by the two methods described is very similar, the automatic extraction seems to produce less-sharp contours, with small areas of "false positive". In general, the automatic method, as observed for segment lengths, tends to overestimate portions of areas of archaeological interest, especially in the case of remains of wall structures that are typically surrounded by large amounts of collapsing material.

In this regard, fieldwork has been very useful both to characterize the different types of archaeological features and to validate the UAV LiDAR-based approach for identifying archaeological features (see Figure 13).

### 3.3. Archaeological Analysis of the Identified Features

The machine learning-based approach, along with data enhancement, enabled to overcome several obstacles in the identification of archaeological features under canopy, including the removal of vegetation (from low to high vegetation), the improvement of visibility of microtopographical variations (Figure 8) related to fossilized urban design of the medieval settlement, and their extraction (Figures 9 and 10).

From a purely archaeological point of view, the analysis of the data acquired yielded much information that could not be acquired by other sensors or methods (Figure 10). The studies on the Perticara site are similar to the result obtained by Masini et al. [4] on Cisterna, a medieval site found in the same geographical context. For the study of the

topographic distribution of the Perticara settlement, the considerable amount of collapse levels obliterating the structures should be taken into account. The degree of visibility of architectural artifacts is dictated by elevation structures and microreliefs, characterized by a convex perimeter and a concave interior. However, the presence of a building can be assumed based on the consistency of the collapsed material in several points.

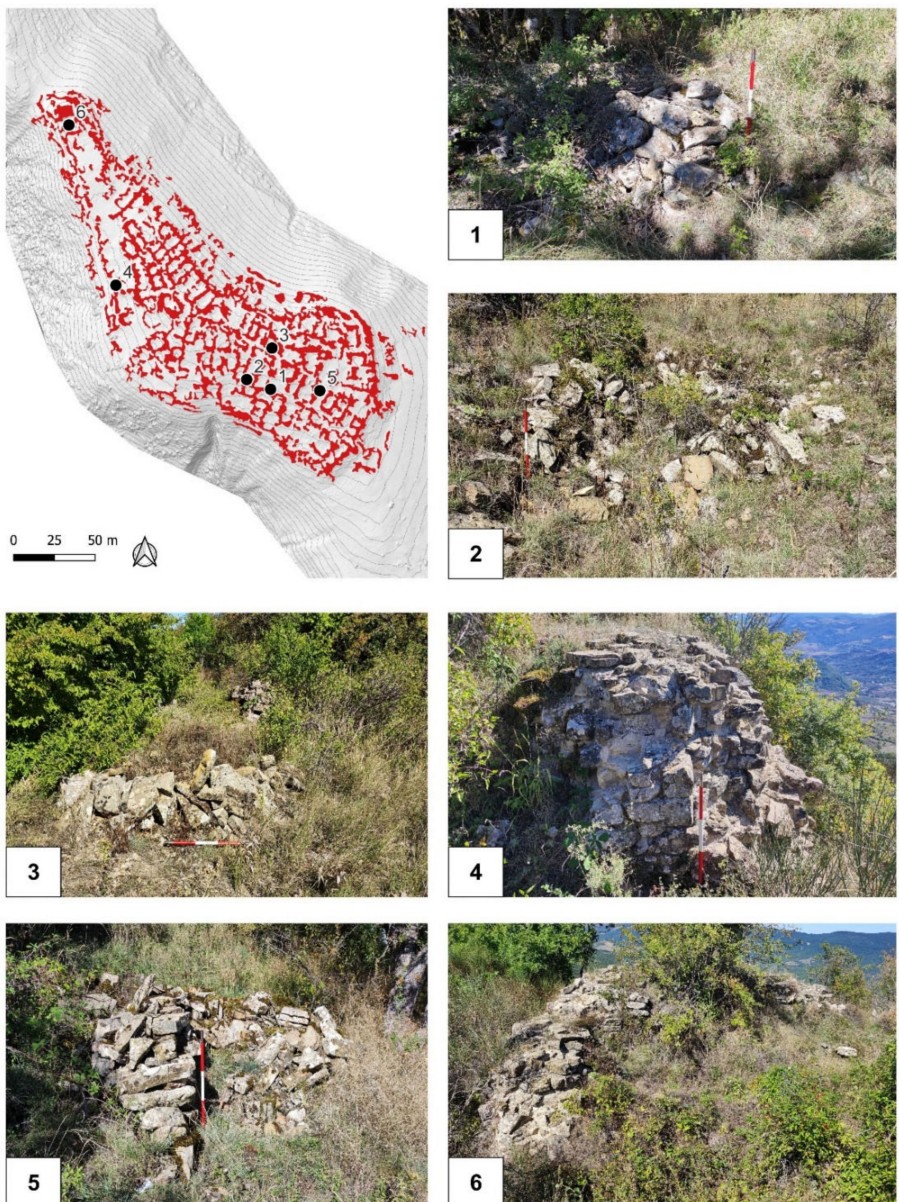

**Figure 13.** Ground data validation: (1–3) remnants of partially preserved collapsed structures and microrelief related to buildings; (4) perimeter walls; (5) partially collapsed structure of which an arch of the first or basement floor can be seen; (6) tower masonry.

Analysis of the derived LiDAR data, as well as of the identified features, shows a highly articulated settlement—confirmed by in situ analyses (see Figure 13)—that is spread over an upland site. The settlement is divided into several sectors, as is often the case with sites from the medieval period. Architectural blocks have been identified within the settlement, which, starting from the top and reaching the lowest parts of the hill, delimit clearly defined built-up areas by their distinct structural and functional characteristics (Figures 14 and 15).

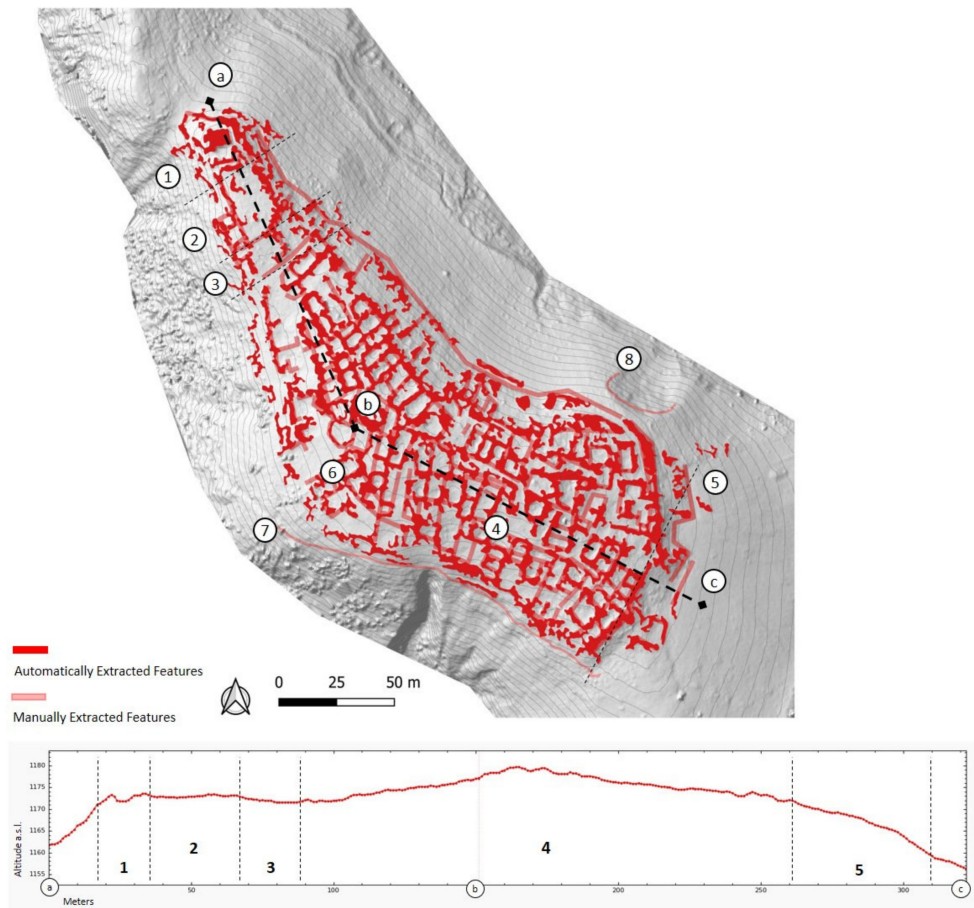

**Figure 14.** Interpretation of the functional blocks of the settlement: (1) tower, (2) parade ground, (3) moat, (4) habitation, (5) defensive perimeter, (6) open square with fountains or cisterns, (7) land-slide cutting through the medieval settlement, and (8) quarry; (a–c) represent the section line of the profile shown below.

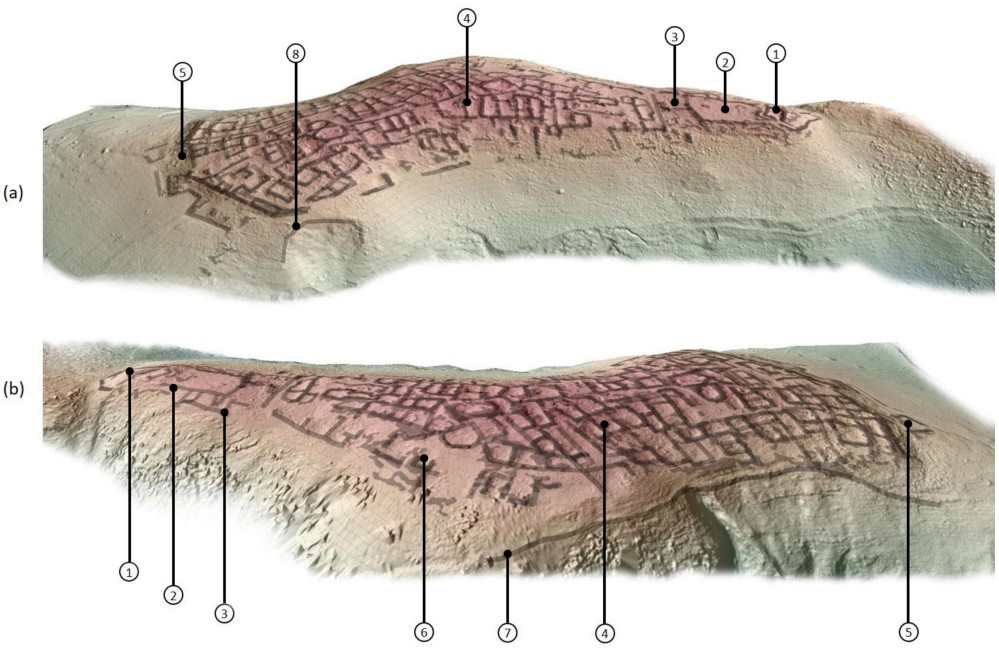

**Figure 15.** DFM hillshading 3D view interpreted feature overlay as in Figure 9: (**a**) northeast view; (**b**) west view.

The site is divided (northeast to southwest) into: tower (9 × 9 m approx.), parade ground (40 m approx.), ditch, built-up area, and defensive perimeter. Also identifiable from the LiDAR data are what appear to be a quarry (Figures 14 and 15, n.8) and the landslide that cut off part of the medieval settlement (Figures 14 and 15, n.7). The habitation consists of quadrangular and rectangular rooms averaging 5 × 5 meters in size, often associated a group of two or more, lacking a probable upper floor. The rooms overlook squares with cisterns or fountains (Figures 14 and 15, n.6) and roads, and there are at least two main road axes, one towards the southern slope and one on the northern slope. One point to note is that the identified habitable areas occupy an area of about 7700 m², which could have reached 15,000 m² or more if one considers that each building had at least one raised floor. This number is consistent with estimates made through archaeological sources that cite a population of about 1000 people (240 hearthstones) in the village during the 13–14 century. The village is spread over several levels and is divided into a number of building cores, with the first located in the lowest area located to the south, composed of rectangular building bodies, some of which are divided into several rooms. A second large nucleus is distributed in the northern and eastern part of the slope and is characterized by an arrangement of terraced building bodies on two different levels. The entrance to the site must have been located to the southeast near the embankment that gives access to the built-up parts (Figures 14 and 15, n.5). The innermost part of the site (northwest) has a fortified area consisting of a moat (Figures 14 and 15, n.3) enclosing a parade ground with several structures that could have functioned as housing for guards, storerooms, or functional places for life in the lordly area. The settlement ends with a quadrangular tower, of which only the lower part is preserved. The settlement of Perticara looks similar to many others from the same period of the Italian Middle Ages [82,83].

## 4. Conclusions

This paper presents a machine-learning approach devised for the LIDAR-based identification of archaeological sites under canopy in hilly regions that pose critical challenges for searching subtle archaeological remains. The presence of dense vegetation and tree cover makes the reconnaissance of archaeological remains very difficult and the erosion, increased by slope, tends to affect over time the microtopographical features of potential archaeological interest, thus making them hardly identifiable. Filtering of LiDAR data, combined with data enhancement methods (e.g., Lee filter, derived LiDAR data, LISA, classifiers, and segmentation), allowed us to overcome several obstacles including (i) removing vegetation, (ii) improving the visibility of features of archaeological interest (Figure 5), as well as (iii) extracting features of archaeological interest (Figures 6 and 7).

Overall, the results of the UAV LiDAR-based approach applied to the Perticara site highlight three important findings from a technological, methodological, and archaeological point of view, as listed below, respectively:

i.　　The resolution of the LiDAR data from the drone is abundantly sufficient to recognize microtopographic features of archaeological interest, even in a context such as Perticara, characterized by such high-wooded cover;

ii.　　The automatic approach of extracting the same features, compared with the qualitative interpretation (in turn corroborated by ground validation), has proven to be effective and therefore mature to be used in operational scenarios of preventive archeology;

iii.　　From an archaeological point of view, the application has allowed the reconstruction of the urban form, and the identification of its constituent elements from a constructive and functional point of view.

**Author Contributions:** N.M. and N.A. coordinated the drafting of the paper. N.M. conceived and directed the investigations. N.A., N.M. and R.L. developed remote sensing-based methodology. N.A. conducted LiDAR data processing. N.A. and V.V. conducted LiDAR data acquisition. The ground truth for the validation of the machine learning approach has been conducted by N.A., A.M.A., V.V., N.A. and F.T.G. Preliminary field survey has been conducted by M.B. (Mario Bentivenga), N.M., M.S., M.B. (Marilisa Biscione) F.C. coordinated the geological study with the contributions of A.M.A., F.T.G. and M.S. The historical research has been conducted by M.B. (Marilisa Biscione) and N.M. The archaeological interpretation has been conducted by N.A., N.M. and V.V. The last version of the article has been revised by N.M. and R.L. All authors have read and agreed to the published version of the manuscript.

**Funding:** The AirLab laboratory has been granted by MUR through the SHINE (Strengthening the Italian Node of E-RIHS) Project (PIR01_00016, PON IR 2014-2020).

**Data Availability Statement:** Data is contained within the article.

**Acknowledgments:** LiDAR investigations in Perticara are part of a methodological development activity of AirLab which is a UAV laboratory of MOLAB of the European Research Infrastructure for Heritage Science (E-RIHS). The AirLab laboratory has been granted by MUR through the SHINE (Strengthening the Italian Node of E-RIHS) Project (PIR01_00016, PON IR 2014-2020).

**Conflicts of Interest:** The authors would like to hereby certify that there are no conflicts of interest in the data collection, processing, and post-processing, the writing and revision of the manuscript, and in the decision to publish the manuscript results.

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
