# Peer review of "UAV LiDAR Based Approach for the Detection and Interpretation of Archaeological Micro Topography under Canopy—The Rediscovery of Perticara (Basilicata, Italy)"

_remotesensing, doi:10.3390/rs14236074_

Round 1

Reviewer 1 Report

This is a very well written paper that details the use of UAV-based LiDAR survey to document a Medieval settlement in southern Italy. The methods are well documented, the paper well organized, and appropriate references appear to be included. Overall, I recommend acceptance of the manuscript. I have only two very small suggestions for improving the paper, which I list below.

Specific comments:

Table 1: Parameters for Openness are missing (it says see above but I don't see any such details elsewhere in the paper). In any case, these details should be provided in the table.

Figure 7: Legend on panels D and E should be fixed. What unit is height measured in? This needs to be specified. Also, I would suggest rounding the numbers to the nearest whole unit and get rid of decimal places. At the very least, limit it to one or two decimals.

Author Response

Please find the responses attached.

Reviewer 2 Report

This paper presents a processing pipeline for UAV-LiDAR for archaeology and an object-based image analysis for semi-automatic detection of standing archaeological features. It also presents an archaeological case study, a medieval deserted village in mountainous context in South Italy.

The article is innovative in some aspects of object-based image analysis. The degree of novelty is moderate, as a very similar pipeline has already been applied to a very similar case study in the same geomorphological and archaeological context by (some of) the same authors. However, this article is important as it demonstrates that object-based image analysis is just as effective and far less laborious to deploy then deep learning approaches that currently dominate in archaeological LiDAR. This alone is reason enough for me to support the publication of this article.

The second innovative aspect is that the article provides new archaeological information for this particular case study. However, this is not discussed in the article, nor is the data disseminated in a reusable format, such as GIS dataset or a small-scale vector plan of the site 

The main problem with the article is that it is not positioned as either a methodological or case study article, but tries to be both. Given some notable gaps in the description of the methodology, I suggest positioning the article as a case study.

In addition, some other shortcomings need to be addressed.

First, I strongly suggest rewriting the entire introduction using the structure of one of the already published workflows/pipelines as a starting point.

Second, some parts of the processing workflow are described in great detail, while others are omitted altogether (e.g., "Finally, the DTM was created..." is used as a placeholder instead of describing the interpolation of the DTM). The methods section (2.3) needs to be rewritten to describe all steps. Alternatively, existing paradata standards can be used (e.g., DOI 10.3390/geosciences11010026, Table 4) and then only selected steps can be described in detail.

Thirdly, some of the most important metadata are missing: 1) the density of the ground point data and/or the DTM confidence map; 2) the description of the archaeological features searched for.

I suggest that the article be rewritten as a case study article that includes a clear description of the processing pipeline and provides archaeological data in a format suitable for reuse by archaeologists.

Attached is a specific list of suggested shortcomings that should be addressed.

Author Response

Please find the responses attached.

Round 2

Reviewer 2 Report

No further comments and suggestions for authors.